# Characterization of Two *Parabacteroides distasonis* Candidate Strains as New Live Biotherapeutics against Obesity

**DOI:** 10.3390/cells12091260

**Published:** 2023-04-26

**Authors:** Bernardo Cuffaro, Denise Boutillier, Jérémy Desramaut, Amin Jablaoui, Elisabeth Werkmeister, François Trottein, Anne-Judith Waligora-Dupriet, Moez Rhimi, Emmanuelle Maguin, Corinne Grangette

**Affiliations:** 1University Lille, CNRS, Inserm, CHU Lille, Institut Pasteur de Lille, U1019-UMR 9017-CIIL-Centre d′Infection et d′Immunité de Lille, 59000 Lille, France; 2Université Paris-Saclay, INRAE, AgroParisTech, Micalis Institute, MIHA Team, 78350 Jouy-en-Josas, France; 3UMR2014-US41-PLBS-Plateformes Lilloises de Biologie and Santé, 59000 Lille, France; 4Université Paris Cité, Inserm UMRS1139, 3PHM, 75006 Paris, France

**Keywords:** *Parabacteroides*, microbiota, live biotherapeutic products, probiotics, obesity, holobiont

## Abstract

The gut microbiota is now considered as a key player in the development of metabolic dysfunction. Therefore, targeting gut microbiota dysbiosis has emerged as a new therapeutic strategy, notably through the use of live gut microbiota-derived biotherapeutics. We previously highlighted the anti-inflammatory abilities of two *Parabacteroides distasonis* strains. We herein evaluate their potential anti-obesity abilities and show that the two strains induced the secretion of the incretin glucagon-like peptide 1 in vitro and limited weight gain and adiposity in obese mice. These beneficial effects are associated with reduced inflammation in adipose tissue and the improvement of lipid and bile acid metabolism markers. *P. distasonis* supplementation also modified the *Actinomycetota*, *Bacillota* and *Bacteroidota* taxa of the mice gut microbiota. These results provide better insight into the capacity of *P. distasonis* to positively influence host metabolism and to be used as novel source of live biotherapeutics in the treatment and prevention of metabolic-related diseases.

## 1. Introduction

The highly diverse microbial community that inhabits the human gut harbors more than 100 trillion microbes, including bacteria, archaea, fungi, protozoa and viruses. Its collective genome, the microbiome, possesses at least 25-times more genes than the human genome [1]. This microbiota lives in a symbiotic relationship with the host and is responsible for multiple physiological functions, notably contributing to the metabolism as well as regulating immune responses [2,3]. There is now clear evidence that the gut microbiome and the associated host–microbiome symbiosis are intrinsically linked with health and diseases. A causal link between the gut microbiota and the development of obesity has been evidenced not only using germ-free mice but also by showing that the transplantation of obesity-associated human microbiota can induce weight gain in lean mice [4,5]. Animal models have confirmed the major role of the gut microbiome in energy extraction from food and energy homeostasis, as well as in regulating the lipid and glucose metabolism [4]. Metagenomics sequencing established that the gut microbiome is dominated by two main phyla, *Bacillota* (*Firmicutes*) and *Bacteroidota* (*Bacteroidetes*), representing 90% of the phylotypes, with the remaining phylotypes distributed between the *Actinomycetota* (*Actinobacteria*), *Pseudomonadota* (*Proteobacteria*) and *Verrucomicrobiota* (*Verrucomicrobia*) phyla [6]. Changes in the gut microbiota composition and function, as well as reduced bacterial diversity, highlight the pivotal role of this microbial ecosystem in the pathogenesis of many chronic disorders, notably in metabolic diseases such as obesity [7], which represents one of the most prevalent problems of public health worldwide. The resulting dysbiosis was suggested to play an important role in the gut barrier dysfunction, allowing the translocation of bacterial products and, thus, the activation of inflammatory pathways. This leads to the appearance of inflamed and dysfunctional adipocytes along with an infiltration of immune cells, notably macrophages, leading to a persistent low-grade inflammation and contributing to the development of metabolic dysfunction and insulin resistance [8,9]. SCFA produced by the healthy gut microbiota through the fermentation of indigestible foods plays a key role in immune and metabolic functions, notably in the control of satiety [10] and in regulating lipid and glucose homeostasis. A decreased abundance of SCFA-producing bacteria observed in patients with obesity has been associated with the alteration of the gut endocrine function [11]. Therefore, there is a growing interest in testing targeted microbiome interventions, notably through the use of selected probiotics. Traditional probiotics (bifidobacteria and lactobacilli) having a long history of use as safe microorganisms and have largely been screened and evaluated in the context of obesity [12]. Many studies, including ours [13,14,15,16,17,18], have reported the beneficial impact of different strains in experimental models with strain-dependent effects and the different mechanisms involved. However, human clinical trials remain scarce [19,20], and validated effects are expected to define the potent strains.

With new advances in metagenomics, microbiome-based therapies are now regarded as new potential strategies for the prevention and treatment of obesity and associated diseases. A growing number of microorganisms that are found more abundant in healthy populations or decreased in patients with obesity, which have also been recognized as beneficial to host health, have been evaluated as next-generation probiotics (NGPs) [7,21,22,23,24,25]. Several strains have been shown to alleviate obesity in experimental models, and some of them, such as *Akkermansia muciniphila*, *Christensenella minuta*, *Eubacterium halii* and *Hafnia alvei* (recently reclassified as *Anaerobutyricum hallii*), have started to be evaluated in clinical trials [26,27,28,29,30,31,32]. Interestingly, pasteurized *A. muciniphila*, which has immunomodulatory properties, was recently recognized as a novel food by the European regulation [33], and *H. alvei* HA4597^®^ strains are now included in a food supplement (EnteroSatys^®^ by TargEDys Company, Lonjumeau, France). Considering the high inter-individual variability in the composition of the gut microbiota and the heterogeneity in the obesity-associated microbial signatures according to different studies, searching for new NGP based on other strategies must be considered [34]. Moreover, it remains essential to better understand the functional properties of specific taxa belonging to the gut microbiota. Few extensive in vitro screenings have been performed to characterize commensal strains isolated from the gut microbiota in comparison to traditional probiotics [14]. We set up a screening approach to evaluate the functional ability of a collection of strains representing different genera and species [35]. This allowed us to characterize bacteria with high potential in the context of IBD, notably three *Parabacteroides distasonis* strains able to counteract colitis in mice [36]. One of these strains (AS93) was also able to induce in vitro the secretion of glucagon-like peptide-1 (GLP-1) [35], an endocrine gut peptide known to play a crucial role in metabolic regulation [37]. We identified promising candidates to use in the context of obesity, notably two *P. distasonis* strains—AS93 and PF-BaE11—exhibiting strong anti-inflammatory activities and the capacity to restore the gut barrier. In the present work, we showed that the two strains are indeed able to induce GLP-1 release in vitro and highlight the promising abilities of the two *P. distasonis* strains to counteract established obesity.

## 2. Materials and Methods

### 2.1. Bacterial Strains and Culture Conditions

*P. distasonis* AS93 was isolated from the feces of heathy human adults, and *P. distasonis* PF-BaE11 was isolated from a fecal newborn sample. Taxonomic assignation was determined based on the Blast comparison of the sequence of the V3-V4 variable region of the 16S ribosomal RNA of the selected strains with the NCBI 16S ribosomal RNA sequences database. The obtained sequences for the AS93 and the PF-BaE11 strains exhibited 99.41% and 99.22% identity with the corresponding sequence of the *P. distasonis* reference strain ATCC 8503, respectively. Bacteria were grown at 37 °C in Brain–Heart Infusion medium Supplemented (BHIS) with 0.5% yeast extract (Difco, Saint Ferréol, France), 0.5 mg/mL hemin (Sigma-Aldrich, Saint Quentin Fallavier, France), 0.5 mg/mL maltose (Sigma-Aldrich), 0.5 mg/mL cellobiose (Sigma-Aldrich), 0.5 mg/mL cysteine (Sigma-Aldrich) and 0.098 mg/L K1 vitamin (Sigma-Aldrich) in an anaerobic cabinet (Jacomex, Dagneux, France) supplied with BIO300 (Air Liquide, Paris, France). After overnight incubation, bacteria were centrifuged at 6000 rpm for 15 min at 4 °C, and bacterial pellets were washed with phosphate-buffered saline (PBS, pH 7.2) in anerobic conditions. For in vivo experiments, dry pellets were frozen in liquid nitrogen and stored at −80 °C. For in vitro experiments, bacterial suspensions (10^9^ CFU/mL) were obtained by adding PBS with 25% glycerol (previously incubated at least 48 h in anaerobiosis) and were frozen in liquid nitrogen before storage at −80 °C.

### 2.2. Measurement of SCFA Production

After measuring the optical density (OD_600nm_) of the overnight bacterial culture, supernatants were collected after centrifugation (15 min at 6000 rpm) and kept at −20 °C. SCFA were measured as previously described [35,38] with slight modifications. Briefly, after the addition of the SCFA standards, each sample was acidified and then extracted using diethyl ether (Biosolve, Dieuze, France), with gentle stirring for 1 h and centrifugation for 2 min at 5000 rpm at 4 °C. The organic layers were derivatized using tert-butyldimethylsilyl imidazole (Sigma-Aldrich), and samples were incubated for 30 min at 60 °C before analysis by gas chromatography-mass spectrometry (GC-MS 7890A-5975C; Agilent Technologies, Montpellier, France) using a 30 m × 0.25 mm × 0.25 µm capillary column (HP1-MS; Agilent Technologies), as previously described [35]. The SCFA concentrations were reported at the mean concentrations divided by the optical density of the culture and expressed as mean ± SEM.

### 2.3. STC1 Enteroendocrine Cell Culture and Induction of GLP-1 Secretion

The ability of the *P. distasonis* strains to induce the secretion of GLP-1 was measured using the neuroendocrine murine cell line STC-1 (ATCC CRL-3254^™^), as previously described [14,35] Cells were grown at 37 °C under 5% CO_2_ in DMEM (Life Technologies, Saint Aubin, France), supplemented with fetal calf serum (10%, Dutscher, Brumath, France), L-glutamine (5 mM) and streptomycin and penicillin (100 µg/mL). For bacterial stimulation, cells were grown in 12-well plates (200,000 cells/well) for 72 h, washed twice with PBS and resuspended in 400 µL of 20 mM Hepes/20 mM Tris pH 7.4 buffer containing 140 mM NaCl, 4.5 mM KCl, 1.2 mM CaCl_2_, 1.2 mM MgCl_2_, 10 mM glucose. Cells were stimulated (or not) with the bacteria (10:1 bacteria/cell) or with butyrate (10 mM final) as a positive control at 37 °C under 5% CO_2_. After 8 h stimulation, the supernatants were collected and centrifuged (10 min at 8000× *g*). DPP-IV enzyme inhibitor (Ile-Pro-Ile, Sigma-Aldrich Germany) was added (100 µM), and samples were stored at −20 °C. The level of active GLP-1 was measured using the V-Plex system and MESO QuickPlex SQ 120 (MesoScale Diagnostics, Rockville, MD, USA).

### 2.4. Murine Model of Diet-Induced Obesity

C57BL/6 JRj male mice (5 weeks old) were purchased from Janvier Labs (Le Genest- St-Isle, France). A high-fat diet (HFD, 45% kcal fat, D12451) and low-fat diet were used as controls, (LFD, 10% kcal fat, D12450B), and irradiated diets were purchased from Research Diets (Brogaarden, Lynge, Denmark). Mice were housed under specific pathogen-free conditions in the animal facility of the Institut Pasteur de Lille (accredited no. A59107) and maintained in a temperature-controlled (20 ± 2 °C) facility with a strict 12 h dark/light cycle and were given ad libitum access to regular chow and water. Housing and procedures complied with current national and institutional regulations and ethical guidelines (Institut Pasteur de Lille/B59-350009 and CEEA 75 Nord-Pas-de-Calais). Experimental protocols were approved by the Ministère de l’Education Nationale, de l’Enseignement Supérieur et de la Recherche, France (accreditation no. APAFIS#2019101811141602).

After a one-week acclimation period, mice were divided into 4 groups (12–13 animals per group, 2 animals per cage) matching the same mean of the starting body weight. Mice were then either fed with LFD diet (control LFD) or HFD diet (obese mice) for 7 weeks. Mice were then gavaged daily (5 days per week) either with 200 µL of buffer (0.2 M sodium bicarbonate, 0.5% glucose; control LFD and HFD mice) or with a suspension of each *P. distasonis* strain (10^9^ CFU in 200 µL buffer) for an additional 9 weeks while maintaining the respective diet (see Appendix A). Body weight and food intake were monitored weekly. An intraperitoneal glucose tolerance (IP-GTT) test was performed as previously described [13] at the 13th week of diet. Blood was collected by retro orbital bleeding of overnight-fasted mice. A solution of glucose (2 g glucose/kg body weight) was administrated by intraperitoneal injection to overnight-fasted mice, and blood glucose levels were recorded 30 min before glucose injection and at different time points after 15, 30, 60, 90 and 120 min using a glucometer (Accu check, Roche, Basel, Switzerland) on blood droplets collected from the tip of the tail vein. After 16 weeks of diet, mice were sacrificed, and subcutaneous (inguinal) (SAT), visceral (epididymal) (EAT) adipose tissues, liver, intestinal segments, fecal samples and cecal contents were collected. Tissue samples were immediately stored in RNAlater^®^ (Ambion, Life Technologies) and frozen at −80 °C.

### 2.5. Histological Analysis of the Adipose Tissue

Adipose tissue samples (EAT) were fixed in 4% paraformaldehyde for 48 h and then dehydrated by increasing concentration of ethanol (from 70% to 100%) and embedded in paraffin. Sections of 5 μm were deparaffinized, rehydrated and stained with hematoxylin and eosin (H&E). Images were acquired using an optical microscope (Axioplan 2 Imaging, Zeiss, Göttingen, Germany). Adipocytes size and distribution were determined by morphometric analysis using ImageJ software (NIH image, National Center for Biotechnology Information). For immunohistochemical staining, EAT tissue sections (5 representative mice per group) were deparaffinized. After a short antigen retrieval (95 °C for 36 min) using Cell Conditioning solution (CC1; Ventana Medical Systems, Illkirsh-Graffenstaden, France) and blocking (BSA 1% for 4 min), slides were incubated with primary antibody against F4/80 (ab100790, diluted 1/100) for 4 h and stained for 30 min with biotin-free HRP multimer (DISCOVERY Ultramap anti-Rb HRP; Roche, Meylan, France) and an HRP-driven chromogen (DISCOVERY ChromoMap DAB Kit; Roche). Slides were counterstained with a modified Mayer’s hematoxylin (Hematoxylin II; Roche) for 4 min, followed by incubation (4 min) with an aqueous solution of buffered lithium carbonate (Bluing Reagent; Roche). A negative control (primary antibody replaced by reaction buffer) and a positive control were included in the staining run. Intensity signals of the immunostaining were measured on a Roche Discovery XT machine and analyzed by a veterinary pathologist (Oncovet-Clinical-Research, Lille, France).

### 2.6. Analysis of Metabolic Parameters

Plasma levels of leptin were measured by an enzyme-linked immunosorbent assay (ELISA) using a specific kit (R&D Systems, Minneapolis, MN, USA). The HOMA-IR index (Homeostatic Model for the Assessment of IR) was then calculated as follows: (fasted serum insulin/fasted serum glucose)/14;1), as previous reported [13]. The concentration of total cholesterol, HDL and VLDL/LDL was measured using quantification kits provided by Abcam (Cambridge, UK). Plasmatic concentrations of fatty acid binding protein (iFABP) were quantified using the iFABP ELISA kit (MyBiosource, San Diego, CA, USA).

### 2.7. Gene Expression Quantification by Real-Time RT-PCR

Tissue samples were homogenized using Lysing Matrix D (MP Bio, Eschwege, Germany) in RA1 buffer (Macherey-Nagel NucleoSpin RNAII isolation kit; Duren, Germany), except for adipose tissue, for which homogenization was performed using TRIzol™ Reagent (Life Technologies Corporation, Carlsbad, CA, USA). Total RNA was prepared using the NucleoSpin RNAII isolation kit (Macherey-Nagel) according to the manufacturer’s recommendation. The quantification and integrity analysis of RNA were performed using a NanoDrop spectrophotometer (260/280 nm, ratio > 2). In addition, cDNA was prepared by the reverse transcription of 1 µg RNA using the high-capacity cDNA reverse transcription kit (Applied Biosystems, Hammonton, NJ, USA). Quantitative reverse transcription PCR (RT-qPCR) was performed using the Power SYBR Green PCR Master Mix (Applied Biosystems, Life Technologies, Warrington, UK) on the QuantStudio™ 12K Flex Real-Time PCR System (Applied Biosystems, Hammonton, NJ, USA). TATA-binding protein (*tbp*) was used as housekeeping gene. Primer sequences used in the study are available upon request. All samples were run in in a single 384-well reaction plate. The relative gene expression (2^−ΔΔCt^) was normalized according to the PCR cycle thresholds (Ct) for the gene of interest and the *tbp* (ΔCt) and to the ΔCt values between HFD/treated and control animal (LFD) groups (ΔΔCt).

### 2.8. Caecal Microbiota Analysis

The genomic DNA from each sample was extracted according to the Godon protocol [39]. The integrity and concentration of the extracted DNA were checked by NanoDrop^®^ spectrometry and electrophoresis. The V3-V4 region was amplified during 30 amplification cycles at 65 °C from the extracted DNA with primers: F343 (CTTTCCCTACACGACGCTCTTCCGATCTACGGRAGGCAGCAG) and R784 (GGAGTTCAGACGTGTGCTCTTCCGATCTTACCAGGGTATCTAATCCT). Amplicon lengths were approximately 450 bp. The amplicon was purified to eliminate non-specific amplifications. Single multiplexing was performed using a homemade 6 bp index, which was added during a second PCR with 12 cycles using the forward primer 5′AATGATACGGCGACCACCGAGATCTACACTCTTTCCCTACACGAC3′ and the modified reverse primer 5′CAAGCAGAAGACGGCATACGAGAT-index-GTGACTGGAGTTCAGACGTGT′3. The resulting PCR products were purified and loaded onto the Illumina MiSeq cartridge according to the manufacturer’s instructions.

Raw sequences were analyzed using the bioinformatics pipeline FROGS (Find Rapidly OTU with Galaxy Solution) [40]. All sequences were first filtered and cleaned to eliminate the one with low sequencing quality for the V3-V4 region using the FROGS-Galaxy tool. The total number of sequences which have been retained was 2,098,238 (92% of sequences). Subsequently, sequences were grouped into operational taxonomic units (OTU) with a level of similarity of 97%. Then, OTUs were filtered by eliminating the chimeric sequences using the vsearch tool [41]. The obtained OTUs were assigned to different taxonomic levels using the Silva database [42].

To investigate microbiota differences between the studied groups, all sequences were rarefied to equal sample depth. Diversity within samples (α- diversity) was assessed using the Observed and Shannon indices. The abundances at the phylum, family and genus scales were calculated by aggregating the OTUs attributed to each studied taxonomic level. For each of these taxa levels, the Kruskal–Wallis statistical test followed by Dunn’s test was performed to detect combinations that were significantly different [43]. Benjamini–Hochberg corrections (BH) were used to avoid false positives (significance threshold = 0.05). The level of significance was set at *p* ≤ 0.05. Statistical analyses were performed using R software (R Core Team, 2015 and 2017 R Foundation for Statistical Computing, Vienna, Austria).

### 2.9. Statistical Analysis

Statistical analyses and graph preparation were performed using GraphPad Prism software (version 7.0 for windows). Comparison between the four groups was performed using non-parametric two-way analysis of variance (ANOVA) followed by Dunn’s multiple comparison post-hoc test. Comparison between two groups was performed by the Mann–Whitney–Wilcoxon test. Data with *p* values ≤ 0.05 were considered statistically significant.

## 3. Results

Several bacterial strains isolated from human feces were previously screened to identify potent live biotherapeutic candidates [35,36]. In this framework, we unraveled the abilities of two *P. distasonis* strains (AS93 and PF-BaE11) to counteract established obesity.

### 3.1. In Vitro Functional Characteristics of the Strains

Regarding the major role played by SFCAs in the regulation of the host metabolism and immune functions, we compared the capacity of the *P. distasonis* strains to elicit their production in basal culture conditions in BHIS medium (Figure 1A). The two strains produced mainly acetate at concentrations reaching 3.2 mM and 2.7 per 1 OD unit for AS93 and PF-BaE11, respectively. They also produced a slight amount of propionate, with AS93 producing a higher amount (0.7 mM/1 OD) than PF-BaE11 (0.2 mM/1 OD).

Knowing the crucial role of glucagon-like peptide-1 (GLP-1) in metabolic regulation, we compared the ability of the *P. distasonis* strains to induce the secretion of this incretin hormone in vitro. The GLP-1 secretion was studied using the murine enteroendocrine cell line STC-1 [44,45]. As previously observed [35], the AS93 strain induced a high and significant (*p* < 0.05) production of GLP-1 (255 ± 59 pM) in comparison to untreated cells, which was higher than the level induced by butyrate, used as positive control (186 ± 11 pM, *p* < 0.05). The level obtained for the PF-BaE11 strain was lower (101 ± 15 pM) but significant (*p* < 0.05) (Figure 1B).

Since the two strains exhibited anti-inflammatory activities and the capacity to induce the secretion of glucagon-like peptide-1 (GLP-1), we evaluated them as promising candidates to use in the context of obesity.

### 3.2. The Two P. distasonis Strains Counteract HFD-Induced Obesity and Adiposity and Improve Glucose Homeostasis

We evaluated the capacity of the two *P. distasonis* strains to counteract established obesity in mice. To this end, mice fed with a high-fat diet (45% fat) for 7 weeks were then orally treated with *P. distasonis* for additional 9 weeks while maintaining the diet (see procedure in Appendix A). The weight gain of the HFD-treated mice was significantly higher (*p* < 0.001) as compared to mice fed with LFD, while it was significantly lower in mice supplemented with the two *P. distasonis* strains (AUC, *p* < 0.05) (Figure 2A). When comparing the weight gain at the beginning of bacterial supplementation, the weight gain of the mice treated with the two strains reached a similar level to what obtained for the lean mice (LFD group) and was significantly lower than in the HFD-fed group (AUC *p* < 0.001 and 0.01 for AS93 and PF-BaE11, respectively, showing ≈ 60–70% reduction (Figure 2B)). Of note, the food intake was not significantly different in comparison to HFD group (Appendix A). As expected, the weight of the visceral and subcutaneous fat depots (EAT and SCAT, respectively) was significantly higher in HFD-fed mice in comparison to lean mice and was significantly (*p* < 0.05) decreased in *P. distasonis*-treated mice (Figure 2C). This was associated with a lower number of large adipocytes and higher number of small ones in the *P. distasonis*-treated groups in comparison to the HFD control group (Figure 2D), confirmed by morphometric analysis (Figure 2E). Obesity is associated with low-grade inflammation, linked to an increased infiltration of macrophages into the adipose tissue [46]. Compared to the HFD-fed mice, the infiltration with macrophages (F4/80 positive cells) was significantly (*p* < 0.01) decreased in the adipose tissue of mice treated with the two strains, the effect being higher with PF-BaE11 (*p* = 0.057; Figure 2F,G). No difference in the weight of the liver (Appendix A) and macroscopic steatosis (not shown) was observed among groups.

The improvement of weight gain and the reduced adiposity after AS93 and PF-BaE11 administration was associated with a slight decrease of fasting glycemia (Figure 3A) and an important reduction in the level of insulin (Figure 3B) and, as such, of the Homeostatic Model Assessment-Insulin Resistance index (HOMA-IR) (Figure 3C) (however, this reduction was not significant). No significant impact on glucose tolerance was observed after a glucose tolerance test (IP-GTT) (Figure 3D). However, both strains were able to limit the level of plasmatic leptin (Figure 3E) and upregulate the expression of the gene encoding the insulin receptor substrate 2 (*irs2*, *p* < 0.01) (Figure 3F). The expression of *Glut4* and *resistin* was decreased in HFD-fed mice, and their expression was restored in mice supplemented with the two strains; however, the difference was not significant (Figure 3G,H).

### 3.3. Supplementation by the Two P. distasonis Strains Decreases Adipose Tissue Inflammation and Improves Markers of Gut Permeability

Targeting the inflammation in adipose tissue, a crucial factor involved in the development of insulin resistance and type 2 diabetes, is an appealing strategy for the treatment of obesity-related metabolic complications [47]. As expected, a significantly increased expression of inflammatory genes (*Tnfa*, *Il17*, *Mcp1*) and gene markers of monocyte-macrophage recruitment (*F4/80*, *Cd11b*, *Cd11c*) was observed in the EAT of obese (HFD-fed) mice (*p* < 0.01 to 0.0001). The administration of the two *P. distasonis* strains significantly decreased the expression of all proinflammatory genes (*p* < 0.05 to 0.001) (Figure 4A), in association with the observed decreased number of infiltrated macrophages (F4/80 immunolabelling). Lipopolysaccharide (LPS) seems to be involved in the transition of macrophages toward the inflammatory phenotype, and the LPS-binding protein (LBP) produced in response to microbial translocation is also a biomarker of obesity-related insulin resistance [48]. The expression of *lbp* was significantly (*p* < 0.01) upregulated in the EAT of obese mice, while it was significantly (*p* < 0.05) downregulated by the two strains (Figure 4A).

At the gut (ileum) level, while we did not observe a significant impact of HFD on the expression of genes encoding the tight junction protein (*Zo1* and *Occludin*), *Zo1* expression was significantly increased in mice treated with AS93 and PF-BaE11 (*p* < 0.05 and 0.01, respectively) (Figure 4B), and *Occludin* expression was restored with AS93 but not significantly. Importantly, while the blood level of iFABP, a marker associated with altered intestinal permeability, was significantly (*p* < 0.05) increased in obese mice, its concentration was decreased in AS93-treated animals in comparison to obese mice (*p* < 0.01; Figure 4C).

### 3.4. P. distasonis Strains Restore Cholesterol and Bile Acids Metabolism and Limit Lipid Metabolism in the Gut

A decreased expression of the genes involved in cholesterol and bile acid metabolism, notably genes encoding the fatty acid binding protein 6 (*Fabp6*), the fibroblast growth factor FGF15 (*Fgf15*), the organic solute transporter alpha (*Slc51a*) and the Niemann–pick C1-like 1 (*Npc1l1*), was observed in the ileum of obese mice; however, the expression was not decreased significantly. Interestingly, both *P. distasonis* strains were able to restore the expression of these genes, notably the expression of *Fgf15,* reported to control the homeostasis of bile acids and to improve glucose homeostasis [49], was significantly (*p* < 0.05) increased after AS93 supplementation (Figure 5A). This was correlated with a slight decrease in total cholesterol, as well as HDL and LDL; however, this decrease was not significant (Appendix A).

Conversely, an increased ileal expression of genes involved in lipid metabolism (*Apoc2*, *p* < 0.05) and transport (*Mttp*, *p* = NS; *Cd36*; *p* < 0.01) was observed in obese (HFD-fed) mice, while their expression was decreased after *P. distasonis* treatment but in a significant manner only for *mttp* by PF-BaE11 (*p* < 0.05) (Figure 5B).

SCFAs are known to play beneficial roles in the regulation of the lipid and glucose metabolism, as well as in the control control of appetite, notably by sensing G-protein-coupled receptors, such as GPR43 and GPR41 [50,51]. Interestingly, we observed that AS93 and PF-BaE11 significantly increased (*p* < 0.05 and 0.01, respectively) the expression of *Gpr41* (Figure 5C).

TGR5, another G-protein-coupled receptor known as a receptor for bile acids as well as other metabolites, is recognized to play a role as a metabolic regulator, notably in energy homeostasis, bile acid homeostasis, as well as glucose metabolism [52]. Although we did not observe significant differences in *Tgr5* expression among groups, PF-BaE11 tended to increase its expression (Figure 5D).

### 3.5. P. distasonis Strains Differentially Modulate the Microbiota

The composition of the cecal microbiota was characterized for the four groups of mice receiving LFD or HFD with or without the supplementation with each of the *P. distasonis* strains (Figure 6). HFD-fed mice exhibited decreased abundances of (i) the *Actinomycetota* phylum (not significant) and the related *Atopobiaceae* family (*p* < 0.05) and (ii) the *Bacteroidota* phlylum (not significant) and the related *Muribaculaceae* family (*p* < 0.01) in comparison to LFD-fed mice (Figure 6A,D). In contrast, the *Ligilactobacillus* genus related to the *Lactobacillaceae* family displayed elevated an abundance compared with the LFD-fed mice’s microbiota (*p* < 0.05, Figure 6D).

Although the two *P. distasonis* strains reduced the alpha diversity of the microbiota as compared to those of HFD-fed mice (*p* < 0.05 and *p* < 0.01 for AS93 and PF-BaE11, respectively, Figure 6B), they did not significantly alter the beta-diversity (Figure 6C). However, the microbiota of LFD-fed mice differed from those of mice under HFD with or without bacteria supplementation. A more detailed microbiota analysis at various taxonomic levels revealed that each *P. distasonis* strain exerted slightly different effects on the mice bacterial taxa (Figure 6A,D).

The cecal microbiota of HFD-fed mice which received the AS93 strain exhibited a lower abundance of *Bacteroidota* phyla (*p* < 0.05) and a lower abundance of the related *Muribaculaceae* family (*p* < 0.001) in comparison to the microbiota of the LFD-fed mice group. In addition, the *Peptococcaceae* (*p* < 0.05), *Peptostreptococcaceae* (*p* < 0.01) and *Lactobacillaceae* (*p* < 0.001) families, all affiliated with the *Bacillota* phyla, exhibited higher abundances than in the LFD-fed mice microbiota. Of note, *Lactobacillaceae* also significantly differed (*p* < 0.05) from the HFD-fed mice microbiota. At the genus level, the abundance of both *Ligilactobacillus* (assigned to the *Lactobacillaceae* family, *p* < 0.05) and *Enterorhabdus* (belonging to the *Eggethellaceae* family, which tended to increase, although not significantly with AS93 supplementation) increased (*p* < 0.05) compared to the microbiota of LFD-fed mice (Figure 6D).

The impact of PF-BaE11 differed from that of AS93. Indeed the abundance of *Actinomycetota* phyla (*p* < 0.05 vs. LFD and *p* < 0.001 vs. HFD), the related *Atopobiaceae* (*p* < 0.01) and *Eggerthellaceae* (*p* < 0.01 vs. LFD group and *p* < 0.05 vs. the HFD group) families and the related *Enterorhabdus* genus (*p* < 0.001 vs. LFD and *p* < 0.01 vs. HFD) increased compared to the microbiota of the LFD- and HFD-fed mice groups, except for the *Atopobiaceae*, which did not significantly differ from the LFD group. The PF-BaE11 supplementation also modified two families and a genus assigned to the *Bacillota* phyla. The *Peptococcaceae* family remained significantly lower than in the HFD group, receiving AS93, while the *Ruminococcaceae* family and the *Ligilactobacillus* genus exhibited lower abundance than in the LFD group (Figure 6D).

## 4. Discussion

Although several health-promoting gut bacteria have been newly identified as promising strains in the control of obesity, the search for new isolates in order to dispose of a large panel of strains able to face the high variability of the resident human microbiome, according to their resistant or permissive level of colonization, remains an important issue [53]. *P. distasonis* belongs to the human core intestinal microbiota, comprising the most common 57 species present in 90% of 184 individuals [6]. These core genera and species are suggested to have important physiological function for the host, and deviations from this core would lead to pathophysiological states, notably obesity [54]. However, contradictory findings have been reported on the beneficial versus the potential pathogenic role of *P. distasonis* species [55]. On the one hand, some studies have reported a negative correlation between *P. distasonis* and weight or waist circumference in humans [56,57]. Reports have described increased levels of this species in diabetic patients, notably with enrichment in the gut microbiota of women with gestational diabetes mellitus [58,59] or a positive correlation with cardiovascular disease (CVD) risk factors, including low-density lipoprotein (LDL), triglycerides (TG), cholesterol, body mass index (BMI) and dyslipidemia [60]. A recent analysis revealed that most potential pathogenic *P. distasonis* strains harbor a single-copy *rfbA* gene encoding the O-antigen, a key virulence molecule of lipopolysaccharides (LPS). These *rbfA* gene variations may help categorize *P. distasonis* strains for pathogenic versus probiotic strain differentiation [61]. On the other hand, a number of reports have highlighted the beneficial effects of *P. distasonis* in metabolic disease conditions. This species has been associated with improved insulin sensitivity in obese subjects [62] in association with therapy outcomes [63]. Some recent studies investigating the impact of metformin on the human gut microbiome have reported an increased abundance of *P. distasonis* in both healthy individuals and newly diagnosed T2D patients, as the most intriguing findings in restoring a healthy condition [64]. Patients following the Mediterranean diet also presented an increased abundance of this species, negatively correlating with waist circumference [62]. An intervention trial giving a multi-fiber bread to 39 individuals at cardiometabolic risk reported a modification of the gut microbiota, mainly a decrease in *Bacteroides vulgatus* and an increase of *P. distasonis* and *Fusicatenibacter saccharivorans* associated with reduced total and LDL cholesterol, insulin and HOMA [65]. A prebiotic intervention trial with nutriose, a resistant dextrin, showed that *P. distasonis* was modulated in the gut microbiota of 28 healthy women involved in this intervention, and the presence of genes encoding a starch-binding membrane complex in given strains amplified the nutriose effect [66]. Interestingly, resistant starch and dextrin were associated with a positive effect on the markers of metabolic disorders, body mass index and fat in healthy people [67]. In experimental models, propolis widely used in traditional medicines for its health-promoting properties, including metabolic regulation and positive anti-diabetic and anti-obesity effects [68], was reported to increase *P. distasonis* abundance in obese mice [69]. Recently, a strain of *P. distasonis* was shown to alleviate obesity and metabolic disorders in mice [70]. *P. goldsteinii* was also shown to prevent body weight gain in obese mice, and protection was associated with increased adipose tissue thermogenesis [71]. Since conflicting reports have underlined the potential health-promoting impact of *P. distasonis* [55], it remains important to better determine the role of this bacterium in human health and evaluate its strain-dependent functional abilities.

We previously conducted an in vitro screening to evaluate the functional capacities of a large collection of commensal strains and highlighted that one-third exhibited multiple putative beneficial properties [35]. Notably, three *P. distasonis* strains exhibited strong anti-inflammatory potential and the ability to strengthen the epithelial barrier and were confirmed to alleviate intestinal inflammation in a murine model of colitis [36]. Recently, 14 *P. distasonis* strains were screened using different in vitro models, allowing the identification of five strains with strong health-promoting functions [72]. In the present study, we observed that two of the *P. distasonis* strains we studied were also able to induce in vitro the secretion of GLP-1, a multifaceted endocrine gut peptide known to play numerous metabolic roles, notably the control of glucose homeostasis [73]. We therefore evaluated their capacity to counteract obesity and showed that oral (intragastric) administration of both strains was highly effective in limiting weight gain and adiposity in obese mice. The protective effect was associated with a significant reduction of inflammatory markers in the adipose tissue, notably a decreased abundance of inflammatory macrophages. This effect should be better deciphered by precisely analyzing the nature of infiltrating cells using flow cytometry. A beneficial effect was also linked to a reduction of leaky gut markers. A decreased expression of genes involved in lipid and bile acid metabolism was also observed. However, the impact on cholesterol levels remains very light. The precise metabolic impact of the strains will be deciphered in the future. Our results globally confirm the report by Wang et al., showing that a *P. distasonis* strain (CGMCC1.30169) isolated from *ob*/*ob* mice that exhibited improvements in metabolic syndrome after treatment with a derivative of a natural compounds was able to limit obesity in *ob*/*ob* and HFD-fed mice, as well as hyperglycemia and hyperlipidemia [74]. The authors also reported that the supplementation by the live murine-originated *P. distasonis* strain did not have a significant effect on the gut microbiota of *ob*/*ob* mice, if not an increase of this taxa. In our work, each of the human-originated strains induced the modification of the mice gut-microbiota, targeting *Actinomycetota*, *Bacillota* and *Bacteroidota* taxa albeit with slight differences. Indeed, AS93 mainly targeted the *Bacteroidota* phylum and two families belonging to the *Bacillota*, while PF-BaE11 only altered the *Actinomycetota* phylum and families. It is noteworthy that in our study, we did not identify an increase in the *Parabacteroides* genus in mice under HFD and treated with AS93 or PF-BaE11. This observation may be related to the variability of microbiota between batches of mice and their permissiveness to colonization by *P. distasonis*. The sensitivity of the strains to the conditions of the intestinal tract could also inactivate some of the gavaged strains. The characterization of the genomes of these two strains appears as a necessary step to identify the functions associated with their adaptation to intestinal conditions, including lipid and bile acid metabolism, as well as possible risk factors, such as the presence of the *rfbA* gene and antibiotic-resistant genes. Wang et al. (2019) established that the metabolic benefit of the *P. distasonis* CGMCC1.30169 strain was mediated by the production of succinate and the modification of the secondary bile acid profile in the gut. They demonstrated that the effects were linked with the activation of the FXR pathway and the promotion of intestinal gluconeogenesis. Our results emphasized that the two *P. distasonis* strains we studied reduced obesity and adiposity, linked notably to the reduced recruitment of macrophages. The strains also improved glucose homeostasis and helped in maintaining gut barrier function. Similar results were obtained with *P. goldsteinii*, which reduced body weight, fat accumulation and markers of inflammation. As we previously reported for the *P. distasonis* strains [36], *P. goldsteinii* also reduced LPS-induced epithelial cells disruption in vitro and improved intestinal permeability in vivo [71]. Altered gut microbiota can indeed lead to gut permeability, notably through the release of components derived from bacterial membranes, and can trigger inflammatory processes.

## 5. Conclusions

Altogether, our results indicate that two strains of *P. distasonis* are able to reduce obesity and associated disorders. Their beneficial impact was notably linked to a decreased inflammation of the metabolic organs and an improvement of the epithelial barrier. This highlights their potential use as new live biotherapeutics to prevent and treat metabolic disorders, even if their precise mechanisms remain to be explored, notably their impact on the gut microbiota composition. However, our study underlines the importance of identifying new commensal strains, notably derived from the core microbiome, and provides new insight on the functional capacities of such bacteria.

## Figures and Tables

**Figure 1 cells-12-01260-f001:**
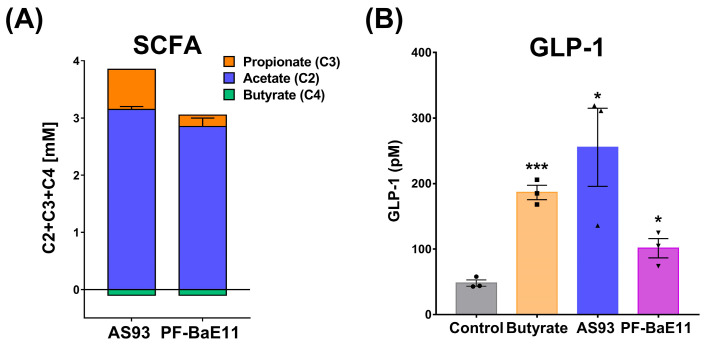
(**A**) Production of acetate, butyrate and propionate measured in the supernatant of bacterial cultures. Results are reported in millimolar per OD_600nm_ unit and expressed as the means of 2 independent experiments ± SEM. (**B**) Production of active GLP-1 in the supernatants of STC-1 cells stimulated for 8 h by the two strains (MOI 10:1) or by butyrate (10 mM) as a positive control. GLP-1 concentrations were measured by Multiplex. ** p* < 0.05 or **** p* < 0.001 in comparison with untreated cells (control).

**Figure 2 cells-12-01260-f002:**
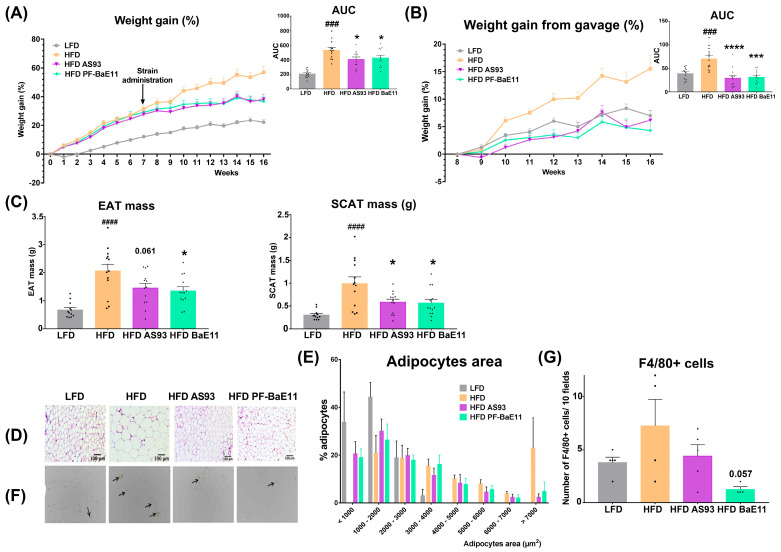
*P. distasonis* AS93 and PF-BaE11 strains limited body weight gain and improved adiposity in obese mice. (**A**) Evolution of the body weight gain expressed in % weight gain from day 0 starting the diet in mice fed an LFD or HFD for 16 weeks and of mice supplemented by the two strains after 7 weeks of HFD (*n* = 12–13 mice per group) and the corresponding AUC (in AU). (**B**) Evolution of the body weight gain expressed as % weight gain from day of starting the bacterial supplementations (week 7) in mice fed an LFD or HFD and HFD-fed mice receiving the two strains (*n* = 12–13 mice per group) and the corresponding AUC (in AU). (**C**) Mass of the epididymal adipose tissue (EAT) and subcutaneous adipose tissue (SCAT) at sacrifice (expressed in g). (**D**) Representative sections of H&E-stained EAT and (**F**) of F4/80 immuno-stained EAT. Black arrows indicate infiltrating F4/80^+^ cells. Scale bars represent 100 μm. (**E**) Distribution of the mean areas of adipocytes in EAT (expressed in %). (**G**) Number of F4/80^+^ cells in EAT (expressed as mean number of positive cells/10 fields, 5 representative mice per group). Results are expressed as mean ± SEM of 12–13 mice per group (5 sections for F4/80 labelling). ^#^ corresponds to regimen effect (HFD versus LFD), * corresponds to supplementation effect (AS93 or PF-BaE11-treted mice versus HFD control mice). * *p* < 0.05; **** p* < 0.001, *^###^* or **** p* < 0.001, *^####^* or ***** p* < 0.0001.

**Figure 3 cells-12-01260-f003:**
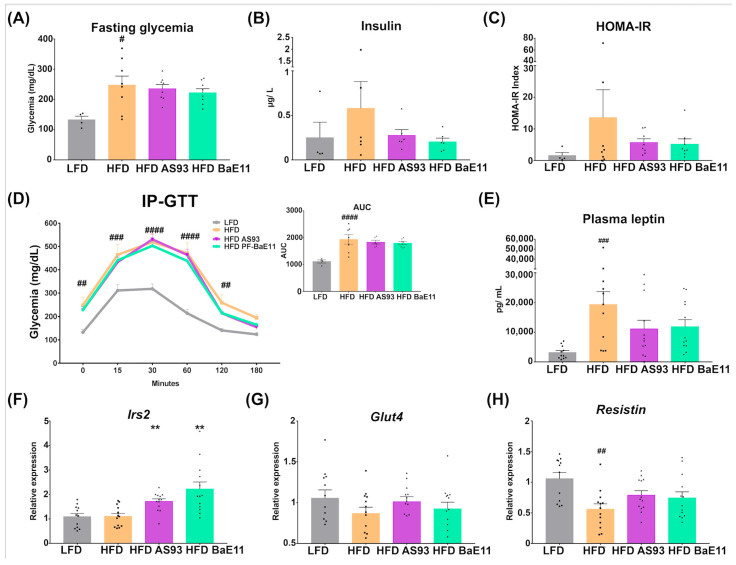
*P. distasonis* AS93 and PF-BaE11 strains improved glucose homeostasis. (**A**) Fasting glycemia (mg/dL), (**B**) plasmatic insulin level (µg/L) and (**C**) HOMA-IR index at sacrifice (after 16 weeks of diet). (**D**) Intra-peritoneal glucose tolerance test (IP-GTT) at 12 weeks of diet and the corresponding AUC (in AU). Blood glucose levels (mg/dL) were measured after glucose IP injection (2 g glucose/kg body weight) at different time points. (**E**) Plasmatic leptin level (pg/mL). Gene expression of (**F**) *Irs2*, (**G**) *Glut 4* and (**H**) *Resistin* quantified by RT-qPCR in the EAT collected at sacrifice. Values are expressed as the relative mRNA levels compared with LFD control mice. Results are expressed as mean ± SEM of 12 mice per group. ^#^ corresponds to diet effect (HFD vs. LFD), * corresponds to the supplementation effects (AS93 or PF-BaE11 vs. Excipient); *^#^ p* < 0.05; *^##^* or *** p* < 0.01, *^###^ p* < 0.001, *^####^ p* < 0.0001.

**Figure 4 cells-12-01260-f004:**
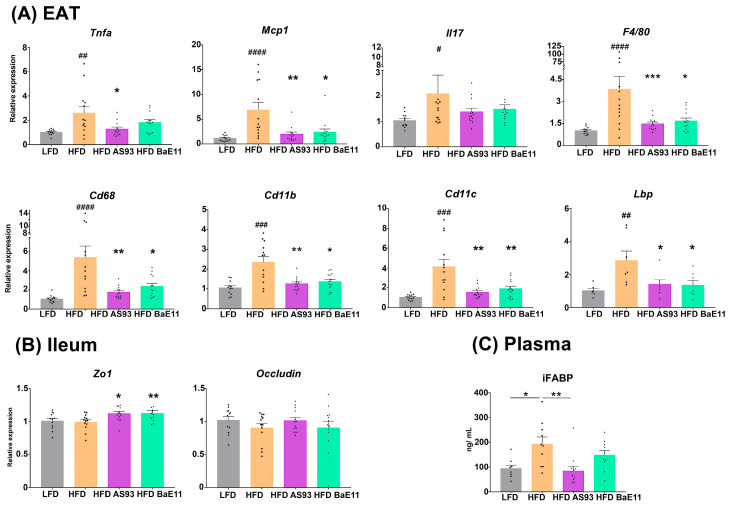
*P. distasonis* AS93 and PF-BaE11 limited the inflammatory gene expression in the EAT and improved gut permeability. (**A**) Expression of inflammatory genes and gene encoding the LPS-binding protein (*Lbp*) in the EAT. (**B**) Expression of genes encoding the tight junction proteins and (**C**) plasmatic levels of iFABP. Values are expressed as the relative mRNA levels compared with LFD control mice. Fatty acid binding protein (iFABP) was quantified by ELISA, and results are expressed in ng/mL. Results represent the means ± SEM of 12 mice per group. ^#^ corresponds to diet effect (HFD vs. LFD), * corresponds to the supplementation effects (AS93 or PF-BaE11 vs. Excipient); ^#^ or * *p* < 0.05; ^##^ or ** *p* < 0.01, ^###^ or *** *p* < 0.001, ^####^
*p* < 0.0001.

**Figure 5 cells-12-01260-f005:**
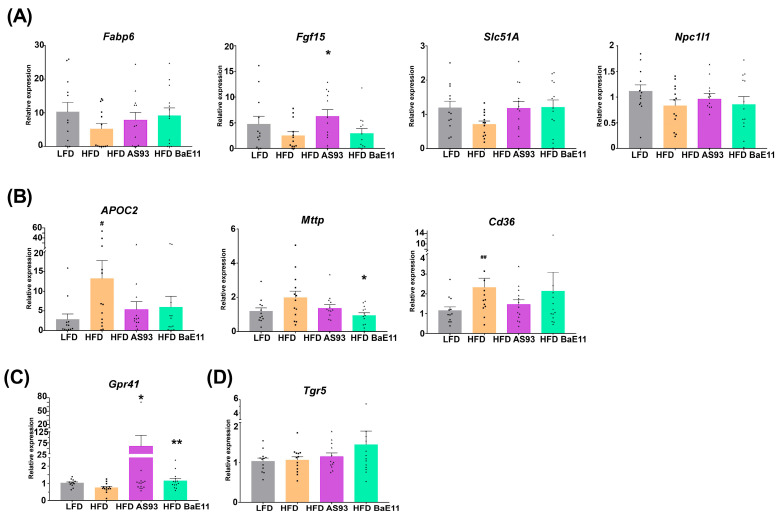
*P. distasonis* AS93 and PF-BaE11 modulated the HFD-induced expression of genes involved in (**A**) cholesterol and bile acid signaling (*Fabp6*, *Fgf15*, *Slc51a* and *Npc1l1*), (**B**) lipid transport metabolism (*Apoc2*, *Mttp* and *Cd36*), (**C**) SCFA (*Gpr41*) or (**D**) bile acids (*Tgr5*) recognition in the ileum. Values are expressed as the relative mRNA levels compared with LFD mice and expressed as means ± SEM of 12-13 mice per group. *^#^* corresponds to regime effect (HFD vs. LFD), *** corresponds to treatment effect (*P. distasonis* vs. Excipient), *** or ^*#*^
*p* < 0.05; **** or ^*##*^
*p* < 0.01.

**Figure 6 cells-12-01260-f006:**
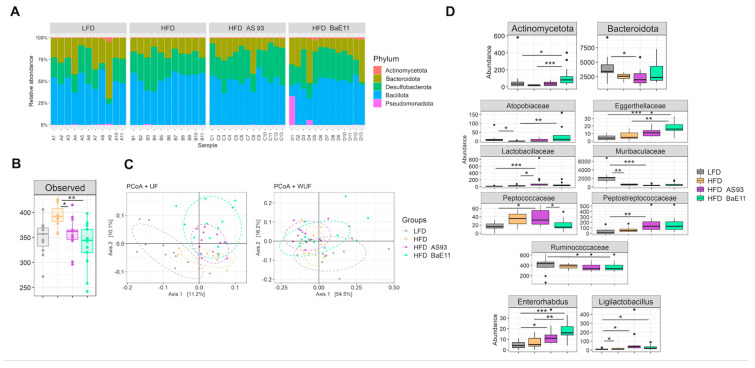
*P. distasonis* AS93 and PF-BaE11 strains differentially modulated the gut microbiota. (**A**) Balance between the main phyla in the cecal microbiota of each mouse. (**B**) Alpha-diversity and (**C**) beta-diversity in the 4 mice groups. (**D**) Bacterial phylum, families and genera displaying differential abundances after comparison between the 4 mice groups. * *p* < 0.05; ** *p* < 0.01, *** *p* < 0.001.

## Data Availability

Not applicable.

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
