# Peer review of "Characterization of Two Parabacteroides distasonis Candidate Strains as New Live Biotherapeutics against Obesity"

_cells, 2023, doi:10.3390/cells12091260_

Round 1

Reviewer 1 Report

This manuscript provided better insight on the capacity 25 of P. distasonis to positively influence host metabolism and to be used as novel source of live bio- 26 therapeutics in the treatment or prevention of metabolic related diseases. It is interesting. However, some issues should be concerned.

1.    These two bacterial strains should have relevant identification data

2.    What is the distribution of these two Parabacteroides distasonis strains in vivo? Whether they will migrate to other tissues rather than simply stay in the gastrointestinal tract.

3.    In the characterization of inflammation in adipose tissue, I suggest to increase the detection of protein expression level, rather than simply rely on gene expression level for characterization. In addition, the infiltration of inflammatory cells also requires immunofluorescence or flow cytometry, which can clearly locate cells.

4.    In Figure 5, since two bacterial strains can regulate the expression of cholesterol related genes, why there is no effect in the regulation of blood lipid expression (Fig. S2)

5.    The definition of Figure 6 is too low to see clearly

Author Response

Please see the attached PDF document.

Reviewer 2 Report

This manuscript is well written, and the reported data are very relavent to the field of gut microbiome-host helath, particular to how P. distansonis affected host metabolism. The manuscript could be further improved and the conclusion could be further supported by providing data of the prevalence and abundances of P. distanonis in the mice modes before and after feeding P. distansomis.

Author Response

Please see the attached PDF document.

Round 2

Reviewer 1 Report

The revised manuscript is better.